# Inducible overexpression of a *FAM3C/ILEI* transgene has pleiotropic effects with shortened life span, liver fibrosis and anemia in mice

**Ulrike Schmidt**[1], **Betül Uluca**[1¤a], **Iva Vokic**[2], **Barizah Malik**[2¤b], **Thomas Kolbe**[3,4], **Caroline Lassnig**[5], **Martin Holcmann**[2], **Veronica Moreno-Viedma**[2], **Bernhard Robl**[2], **Carina Mühlberger**[6], **Dagmar Gotthardt**[6], **Maria Sibilia**[2], **Thomas Rülicke**[7], **Mathias Müller**[5], **Agnes Csiszar**[2]*

**1** IMP—Research Institute of Molecular Pathology, Vienna, Austria, **2** Center for Cancer Research, Medical University of Vienna, Vienna, Austria, **3** Biomodels Austria, University of Veterinary Medicine Vienna, Vienna, Austria, **4** Department IFA-Tulln, University of Natural Resources and Life Sciences, Vienna, Austria, **5** Institute of Animal Breeding and Genetics, University of Veterinary Medicine Vienna, Vienna, Austria, **6** Institute of Pharmacology and Toxicology, University of Veterinary Medicine Vienna, Vienna, Austria, **7** Department of Biomedical Sciences, University of Veterinary Medicine Vienna, Vienna, Austria

¤a Current address: Department of Molecular Biotechnology, Turkish-German University, Istanbul, Turkey
¤b Current address: School of Biochemistry and Biotechnology, University of the Punjab, Lahore, Pakistan
* agnes.csiszar@meduniwien.ac.at

**Data Availability Statement:** All relevant data are within the paper and its Supporting Information files.

## Abstract

FAM3C/ILEI is an important factor in epithelial-to-mesenchymal transition (EMT) induction, tumor progression and metastasis. Overexpressed in many cancers, elevated ILEI levels and secretion correlate with poor patient survival. Although ILEI's causative role in invasive tumor growth and metastasis has been demonstrated in several cellular tumor models, there are no available transgenic mice to study these effects in the context of a complex organism. Here, we describe the generation and initial characterization of a Tet-ON inducible *Fam3c/ILEI* transgenic mouse strain. We find that ubiquitous induction of ILEI overexpression (*R26-ILEI^ind*) at weaning age leads to a shortened lifespan, reduced body weight and microcytic hypochromic anemia. The anemia was reversible at a young age within a week upon withdrawal of ILEI induction. Vav1-driven overexpression of the *ILEI^ind* transgene in all hematopoietic cells (*Vav-ILEI^ind*) did not render mice anemic or lower overall fitness, demonstrating that no intrinsic mechanisms of erythroid development were dysregulated by ILEI and that hematopoietic ILEI hyperfunction did not contribute to death. Reduced serum iron levels of *R26-ILEI^ind* mice were indicative for a malfunction in iron uptake or homeostasis. Accordingly, the liver, the main organ of iron metabolism, was severely affected in moribund ILEI overexpressing mice: increased alanine transaminase and aspartate aminotransferase levels indicated liver dysfunction, the liver was reduced in size, showed increased apoptosis, reduced cellular iron content, and had a fibrotic phenotype. These data indicate that high ILEI expression in the liver might reduce hepatoprotection and induce liver fibrosis, which leads to liver dysfunction, disturbed iron metabolism and eventually to death. Overall, we show here that the novel Tet-ON inducible *Fam3c/ILEI*

**Funding:** The study was supported by a grant of the Austrian Science Fund (FWF) ZK-81B (DG and CM), the Overseas Scholarship of the University of the Punjab, Pakistan (BM) and the Fellinger Krebsforschung (AC). The funders had no role in study design, data collection and analysis, decision to publish, or preparation of the manuscript.

**Competing interests:** The authors have declared that no competing interests exist.

transgenic mouse strain allows tissue specific timely controlled overexpression of ILEI and thus, will serve as a versatile tool to model the effect of elevated ILEI expression in diverse tissue entities and disease conditions, including cancer.

## Introduction

The FAM3 family of cytokines were first identified from a search of genomics databases for molecules predicted to be structurally similar to interleukins and other related cytokines with a four-helix-bundle [1]. The search found four genes *FAM3A*, *FAM3B*, *FAM3C*, and *FAM3D* with a hydrophobic leader sequence [1]. The protein expressed by *FAM3C* was later shown in the context of cancer to be involved in epithelial-to-mesenchymal transition (EMT) [2]. Based on this action and the presumed similarity to interleukins FAM3C was renamed interleukin-like EMT inducer (ILEI). However, subsequent crystal structure analysis revealed that the FAM3 family in fact constitutes a distinct class of signaling molecules [3, 4], FAM3C/ILEI being unique in the family due to its covalent dimerization properties [3, 5].

Human tumors from various cancers overexpress ILEI with altered subcellular localization, which is associated with changes in the secretion levels of the protein [6], elevated ILEI levels also linked to gene amplification [7]. EMT converts adherent epithelial tumor cells into highly invasive mesenchymal cells [8]. ILEI has been demonstrated to induce invasion in cancer cell models and metastasis from xenografts [9–11]. Additionally, altered ILEI localization has been linked to metastasis and survival in human breast and hepatocellular carcinomas [9–11], and high levels of ILEI expression have correlated with poor prognosis in colorectal cancer [12]. Despite a growing understanding of the role of elevated ILEI levels in causing invasive tumor growth and metastasis, methods available to study these effects in the context of a complex organism are limited.

ILEI has also been implicated in other diseases. Low levels of ILEI expression might be a factor in Alzheimer's disease development [13]. While ILEI may also drive EMT in renal fibrosis [14]. *FAM3C* genetic variants have been associated with bone mineral density, dyslipidemia and lipid traits, and sudden sensorineural hearing loss [15–17]. Therefore, mouse models have been developed to investigate non-cancer effects of ILEI. Two previous studies have described a *Fam3c* knockout mouse strain, with a focus on bone mineral density [18, 19]. These models demonstrated relatively minor effects on the mouse phenotype. Another study evaluated the overexpression of human *FAM3C* in neuronal cells under the control of the mouse prion promoter to understand the mechanism of Alzheimer's disease [20]. However, both of these approaches lack the option to manipulate ILEI expression in time and in specific tissues of interest. The Tet-On system is a dual transgenic system involving a reverse tetracycline-controlled transactivator (i) that only recognizes the tetracycline-responsive promoter element (TRE) (ii) in the presence of doxycycline (Dox) [21, 22]. Therefore, using this system to produce transgenic mice can provide reversibly inducible gene expression in a tissue specific manner. A safe consideration for genes that are likely to have lethal or highly debilitating effects. This system has been used successfully to produce mouse models for the study of many different genes [23–26].

In this study, we describe the generation and initial characterization of a Tet-On inducible *Fam3c/ILEI* transgenic mouse strain. This model will provide a useful tool in understanding the complex mechanisms of ILEI action in the context of a whole organism in a wide range of cancers and other diseases.

## Materials and methods

### Ethics statement

Mice were housed under barrier conditions in specific pathogen-free quality according to FELASA recommendations [27]. All animal experiments were discussed and approved by the Ethics and Animal Welfare Committee of the University of Veterinary Medicine Vienna, the Research Institute of Molecular Pathology, and the Medical University of Vienna and conform to the guidelines of the national authority (the Austrian Federal Ministry for Science and Research) as laid down in §8ff of the Animal Science and Experiments Act (Tierversuchsgesetz–TVG; refs BMWF-68.205/0204-C/GT/2007; BMWF-68.205/0210-II/10b/2009, MA58/001489/2008/12, BMWF-66.009/0065-II/10b/2009, BMWFW-66.009/0319-V/3b/2019) and according to the guidelines of FELASA and ARRIVE. A humane endpoint was conducted by cervical dislocation if death of the animals was to be expected during the following hours. As pale skin coloration and weight loss/retarded growth accompanied the overall phenotype, palpable hypothermia and unresponsive condition were used as the most reliable criteria to determine moribundity.

### Cloning, gene targeting, and mice

For the establishment of *ILEI*ind mice we used a 2nd generation Tet-On system based on site-specific recombination in embryonic stem (ES) cells [28, 29]. We targeted a construct carrying the Tet operator (tetO) followed by a FLAG-tagged *Fam3c* cDNA (Genbank accession number NM_138587.4) into the *Col1a1* locus of KH2 ES cells by Flp/frt-mediated recombination as described earlier [30]. Briefly, *Fam3c* coding sequence bearing a FLAG tag at its 3' end was re-cloned from the pcDNA3.1-ILEI-FLAG construct [6] via EcoRI (Thermo Scientific) into the vector pBS319-RGBpA directly under the control of a doxycycline-inducible promoter element [28] and designated pBS31-ILEI-FLAG. The transgenic vector and a vector encoding the Flp recombinase (pCAGGS-FLP [28]) were co-transfected into KH2 ES cells. ES cell clones with correct recombination were selected by hygromycin B, correct and single integration of the construct was verified by Southern blot analysis as described earlier [30]. Tight and inducible regulation of gene expression was confirmed by *in vitro* Dox induction tests. Selected ES cell clone #7 was injected into C57BL/6N blastocysts and the resulting chimeras were crossed to B6N;129Sv mice. Dual-transgenic progenies heterozygous for the *M2rtTA* in the *Rosa26* locus and the dox-inducible ILEI-FLAG construct in the *Col1a1* locus were intercrossed to obtain homozygous dual-transgenic B6N;129Sv-Gt(ROSA)26Sortm1(rtTA*M2)Jae-Col1a1tm1 (tetO-ILEI-FLAG)Biat mice (referred to here as *R26-ILEI*ind mice). In parallel, mice were backcrossed to C57BL/6J mice by speed congenic procedure [31] and crossed to *Vav-rtTA3* mice [32] (referred to here as *Vav-ILEI*ind mice). All experiments were performed using *R26-ILEI*ind male mice in a B6N;129Sv and *Vav-ILEI*ind male and female mice in a C57BL/6J genetic background. Genotyping of *R26-ILEI*ind mice and cells was performed as described earlier [30]. The *Vav-rtTA3* transgene was genotyped as previously described [32]. Dox was supplied to *R26-ILEI*ind mice via drinking water (1 mg/ml, supplemented with 5% sucrose) and to *Vav-ILEI*ind mice in the diet (1000 mg/kg, ssniff). Animals were checked daily to monitor their health status by assessing nesting performance and examining groomed fur and posture. Autopsy was performed on rare moribund mice and on all surviving animals at around day 240 of life, which was set as the end point of the experiment. During macroscopic post-mortem examinations, we did not detect any spontaneous cancer formation.

## Blood tests

Whole blood was collected weekly using K3EDTA MiniCollect tubes (Greiner Bio-One) by bleeding mice on the tail vein. White blood cell count (WBC), red blood cell count (RBC), hemoglobin (HGB) and hematocrit (HCT) levels were measured using an animal blood counter (sciI Vet abc). Blood sera were collected by centrifugation, diluted 1:2–4 upon need in isotonic saline solution and used to determine aspartate aminotransferase (AST), alanine transaminase (ALT), and serum iron levels using Hitachi Cobas C111 and Hitachi Cobas C311 (Roche) devices, respectively.

## Cell culture

ES and bone marrow cells were cultivated as previously described [33]. For *in vitro* induction experiments, cells were treated with doxycycline-hydrochloride (Sigma) dissolved in water at concentrations ranging from 0.2–5 µg/ml.

## Western blot analysis

Protein lysates and western blots were performed as described [6] using FLAG (Sigma), ILEI [11], Erk1, beta-actin and vinculin (Cell Signaling Technologies) primary antibodies and horseradish peroxidase (HRP)-coupled secondary antibodies (Jackson ImmunoResearch).

## Immunohistochemistry, prussian blue, and picrosirius red staining

Primary antibodies against ILEI [11], FLAG (Sigma), Ki67 (Abcam), activated caspase 3 (act-Casp3, Abcam), fibronectin, E-cadherin, and S100A4 (Cell Signaling Technologies) were used for immunohistochemistry by applying standard protocols. Briefly, 4-µm sections of paraffin embedded mouse tissues were deparaffinized, antigen retrieval was performed in citrate buffer (Dako). After peroxidase treatment and blocking (2% BSA, 10% horse serum and 0.1% Tween 20 in PBS), slides were incubated overnight at 4˚C with primary antibodies diluted in blocking buffer, followed by washing and then incubation with HRP-conjugated SignalStain Boost immunohistochemistry detection reagent (Cell Signaling Technology) for 30 minutes at room temperature, developed with DAB substrate (Dako) followed by hematoxylin counterstain, rehydration and mounting. Hematoxylin and eosin (H&E) staining (Sigma-Aldrich) staining was performed according to standard protocols. Prussian blue staining for ferric iron detection was performed as described earlier [34, 35]. Briefly, equal parts of 5% potassium ferrocyanide (Merck) and 5% hydrochloric acid (Sigma) were mixed and heated until vaporization in a microwave. Sections were immersed for 8 minutes in the heated solution and counterstained with nuclear fast red (Sigma) for 7 minutes, followed by washing in tap water, dehydration and embedding. Histological visualization of collagen was performed using the Picric-Sirius Red Stain Kit (Scy-Tek) according to the manufacturer's instructions. Slides were digitized with a Pannoramic SCAN II slide scanner (3DHistech) in extended focus scanning mode using a 20X plan-apochromat objective (0.8 NA) and a 5Mpxl sCMOS camera. Quantification was performed using automized histology quantification software (Definiens Tissue studio® 4.3) and in case of Prussian blue staining by manual counting of "blue" (ferric iron-containing) hepatocytes in randomly selected areas using QuPath 0.3.2 [36].

## Flow cytometry of blood and spleen

The cellular fraction of 100µl freshly isolated blood was suspended in 1x PBS and transferred into 3ml erythrocyte lysis buffer (0.15 M NH4Cl, 10 mM KHCO3, 0.1 mM Na2EDTA, pH 7.2–7.4). After lysis leukocytes were pelleted, washed, and blocked with anti-CD16/32 antibody

(BioLegend), followed by staining with the following fluorophore conjugated primary antibodies for 30 min at 4°C: CD11b, Gr1, CD19 and CD3ε (Biolegend). Single-cell suspension of freshly isolated splenocytes were prepared as described earlier [37] and subsequently blocked with anti-CD16/32 antibody (BioLegend). Splenocytes were stained with following fluorophore conjugated primary antibodies for 30 min at 4°C: Ly6G, CD11b, CD11c, CD19, F4/80, MHC class II, Ly6C and CD3ε (Biolegend). After incubation, the samples were washed, filtered, and stained with SYTOXgreen viability dye (Thermo Fisher) according to the manufacturer's recommendation to exclude dead cells. Leukocytes and splenocytes were recorded by a FACSCanto (BD Bioscience) and by a Fortessa (BD Bioscience) cytometers, respectively and analyzed by FlowJo v10 software.

## Statistical analysis

Statistical analysis was done with GraphPad Prism 8.0. For comparison of one parameter between two groups we used an unpaired two-tailed student's $t$-test, comparison across multiple groups was performed by one-way analysis of variance (ANOVA) with *Bonferroni* posthoc analysis for multiple comparisons to determine statistical significance or in specific cases with student's $t$-test with pairwise comparisons. Survival analysis was performed using Kaplan-Meyer plots and log rank statistics. p-values of lower than 0.05 were considered statistically significant (*$p<0.05$, **$p<0.01$, ***$p<0.001$). Error bars are represented as standard error of mean (SEM) of at least three independent biological replicates.

## Results

### Generation of KH2-ILEI-FLAG ES cells with inducible expression of ILEI-FLAG

The first step in generating the Tet-On inducible *Fam3c/ILEI* transgenic mouse strain was to establish KH2 ES cells expressing Dox-inducible FLAG-tagged ILEI. As shown in Fig 1A, we used the flp-in system to produce KH2 embryonic mouse stem cells expressing the M2 reverse tetracycline controlled transactivator (M2rtTA) which recognizes the tetO sequence of the Tet response element (TRE) on the *Fam3c/ILEI-FLAG* transgene only in the presence of Dox. Transient transfection confirmed that the cloned tet:ILEI-FLAG construct resulted in the expression of a FLAG tagged protein with the expected 25 kDA size only upon induction with Dox (Fig 1B). Single cell clones of the flp-in genomic targeting were selected with antibiotics and further sub-selected after confirming targeted insertion of the transgene by PCR (Fig 1C) and FLAG western blot analysis to verify protein expression of correct size and tight regulation of Dox inducibility (Fig 1D). Southern blot analysis of genomic DNA with a probe specific for the engineered locus was performed to validate single copy insertion of the transgene (Fig 1E). Finally, clone #7 was selected for blastocyst injection.

### The *Fam3c/ILEI-FLAG* transgene is expressed in a broad range of tissues and organs of *R26-ILEI^{ind}* mice upon doxycycline induction

Blastocyst injection gave rise to 8 male mice with 30–50% chimerism and one of them showed germ-line transmission generating one founder male mouse. This founder was crossed to C57BL/6N mice and the first offspring generation was analyzed for transgene expression (Fig 2). First, the expression of the Tet-inducible *Fam3c/ILEI-FLAG* transgene was analyzed in *ex vivo* bone marrow culture after treatment with Dox for 24- or 48-hours at different concentrations (Fig 2A). Western blots of bone marrow isolated from dual-transgenic *Rosa26rtTA-I-LEI^{ind}* (*R26-ILEI^{ind}*) mice showed ILEI-FLAG protein expression upon Dox administration to

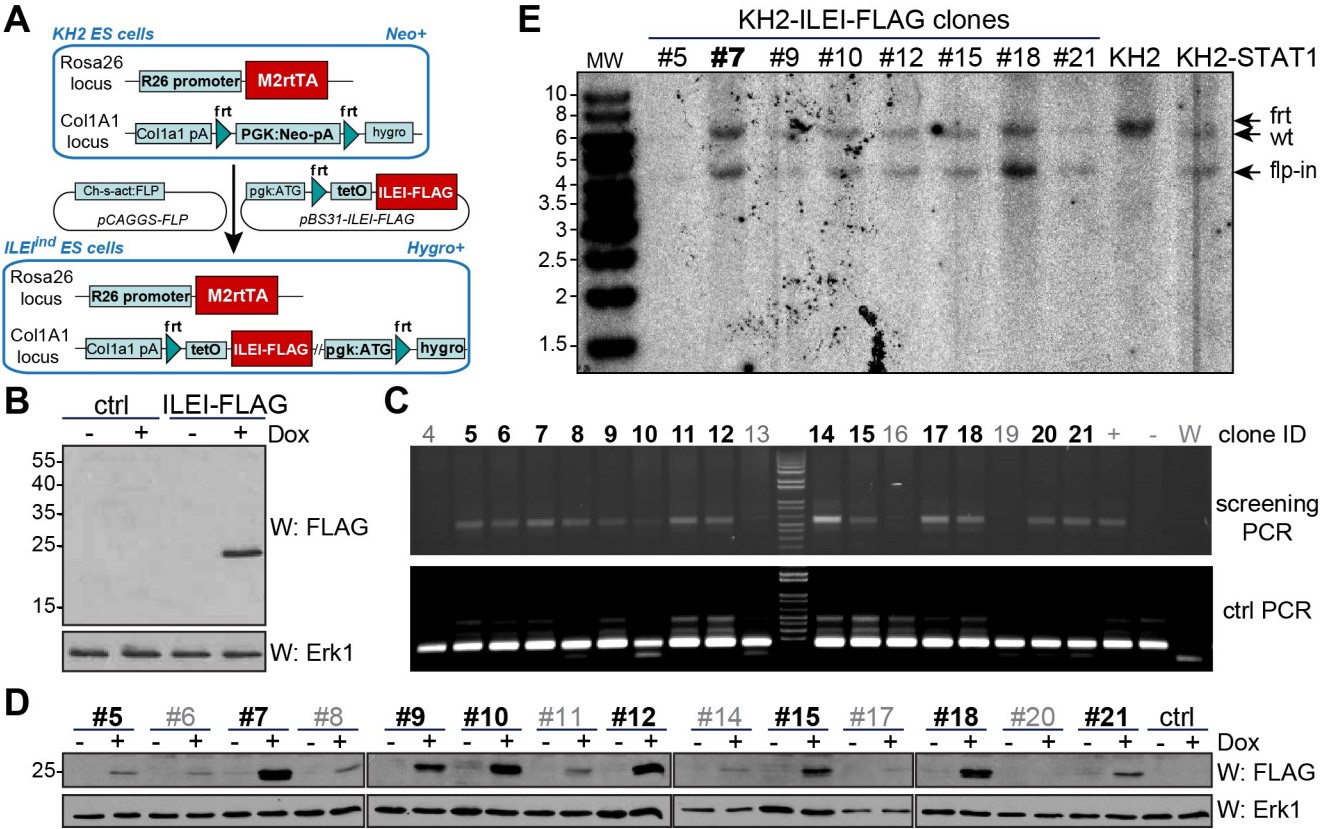

**Fig 1. Generation of KH2-ILEI-FLAG ES cells.** (A) Scheme of the flp-in targeting strategy in KH2 ES cells. Adapted from [30]. (B) FLAG Western blot analysis of KH2 ES cells after transient transfection with the *Fam3c/ILEI-FLAG* transgene targeting construct with or without Dox induction. Erk1 was used as loading control. (C) Validation of transgene targeted insertion by PCR on single clones of KH2 ES cells after Hygromycin selection. Clone IDs in black were selected for further analysis. (D) FLAG western blot analysis of sub-selected KH2-ILEI-FLAG clones with or without Dox induction. Erk1 was used as loading control. Clone IDs in black were selected for further analysis. (E) Southern blot analysis of sub-selected KH2-ILEI-FLAG clones with a probe specific for the engineered locus. KH2 cells were used as negative and KH2-STAT1 cells [30] as positive controls. Clone #7 in bold was selected for blastocyst injection.

the cell culture, while bone marrow derived from a control *Rosa26rtTA* mouse did not. Higher dose of Dox induced higher levels of protein expression (Fig 2A). To evaluate inducibility of ILEI expression *in vivo*, ILEI protein expression levels were analyzed in various organs from mice kept on normal or Dox-containing drinking water for three days before sacrifice (Fig 2B and 2C). Protein extracts of spleen and bone marrow, as well as blood sera were examined by western blot (Fig 2B). The results showed low levels of endogenous mouse ILEI expression in splenocytes of all mice, while samples from the mice on Dox water had obviously increased ILEI protein expression. There was no endogenous ILEI expression in the bone marrow or blood sera, but mice receiving Dox showed high levels of the ILEI protein in these samples (Fig 2B). Histology sections of the intestines, kidney, liver, lung and skin also showed that mice receiving Dox expressed elevated levels of ILEI-FLAG in these organs as indicated by increased staining intensity for ILEI, while those not receiving Dox showed little or no ILEI expression (Fig 2C, left panel). *De novo* transgene expression was verified by immunohistochemistry against the transgene-specific epitope tag FLAG (Fig 2C, right panel). Of note, transgene expression was not detectable in the single-layer alveolar epithelium of the lung, but in other lung structures. These results show that transgenic *Fam3c/ILEI-FLAG* was efficiently expressed across a wide range of tissues and organs in the *R26-ILEI^{ind}* mice and transgene expression was detected only after induction with Dox.

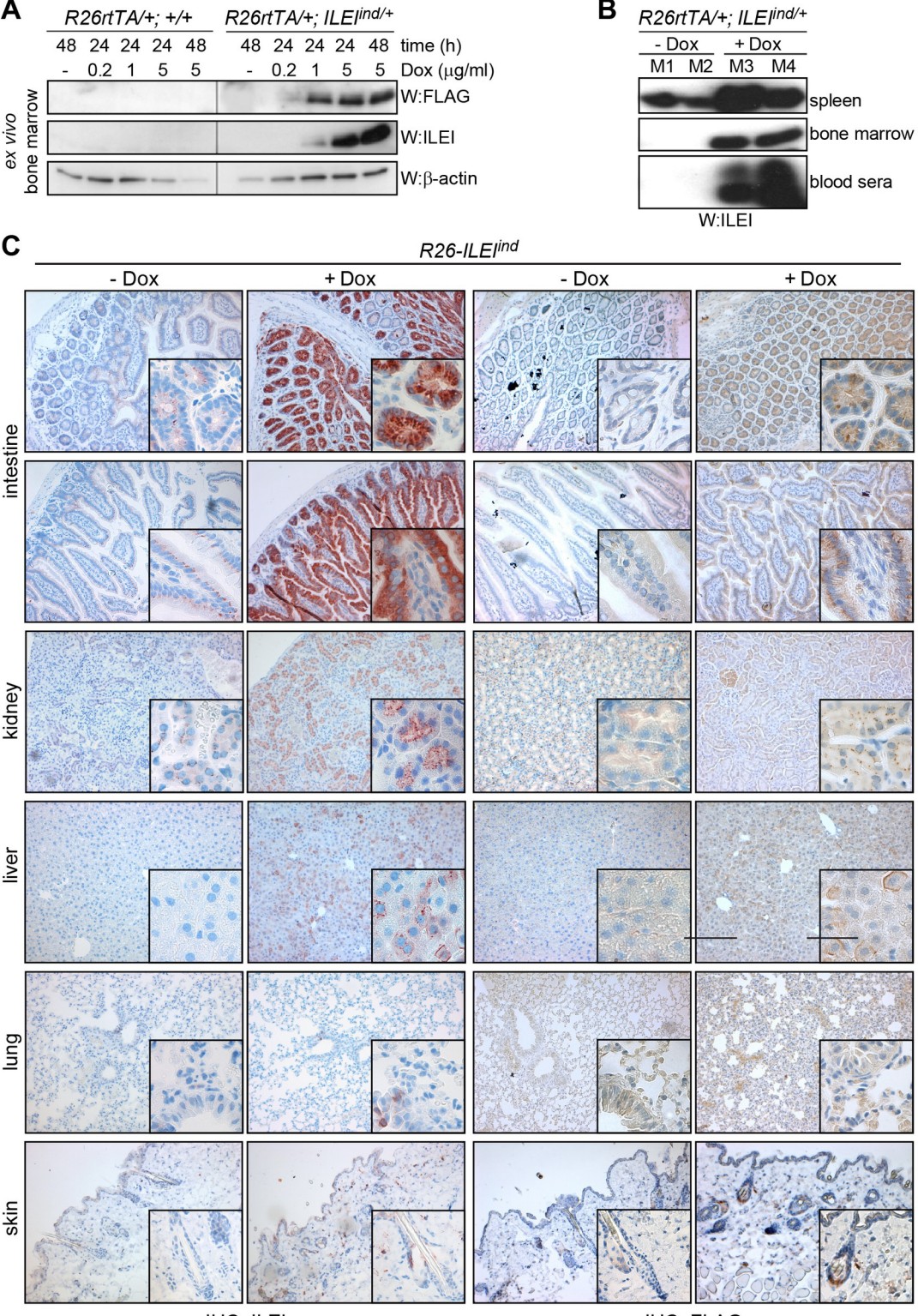

**Fig 2. The *Fam3c/ILEI-FLAG* transgene is broadly expressed in *R26-ILEI^ind* mice upon doxycycline induction.** (A) Western blot analysis of bone marrow isolated from control *Rosa26rtTA* and dual-transgenic *Rosa26rtTA-ILEI^ind* (*R26-ILEI^ind*) mice after *ex vivo* culture for 24 or 48 hours in the presence of the indicated Dox concentrations. FLAG and ILEI antibodies were used for transgene detection, beta-actin was used as the loading control. (B) ILEI western blot analysis of spleen, bone marrow, and blood sera freshly isolated from *R26-ILEI^ind* mice kept on normal or switched to Dox drinking water (1 mg/ml Dox and 5% sucrose) 3 days before sacrifice. Please note, splenocytes also showed endogenous ILEI expression. (C) ILEI and FLAG immunohistochemistry on thin sections of the intestine, kidney, liver, lung, and skin of *R26-ILEI^ind* mice kept on normal or switched to Dox drinking water (1 mg/ml Dox and 5% sucrose) 3 days before sacrifice. Scale bar, 50μm; scale bar for inlets, 20μm.

## ILEI overexpression induced at weaning age results in a reduced life span with reduced body weight and reversible microcytic hypochromic anemia

To analyze the effects of ubiquitous ILEI overexpression *in vivo* we switched mice to Dox-supplied water at weaning age and monitored phenotypic changes and their overall fitness over time. *R26-ILEI^ind* mice receiving Dox drinking water from 3 weeks of age showed a reduced life span compared to *R26-ILEI^ind* mice without Dox or mice with Dox but bearing the *ILEI^ind* transgene alone. *R26-ILEI^ind* mice receiving Dox died with 60% penetrance with a median survival of 220 days (Fig 3A). There was a significant decrease in body weight of *R26-ILEI^ind* mice on Dox water compared to *R26-ILEI^ind* mice without Dox ($p<0.0001$) and *ILEI^ind* mice with Dox ($p<0.0001$) being primarily penetrant in moribund *R26-ILEI^ind* mice on Dox (Fig 3B). On the one hand, blood analysis indicated little effects on WBC counts and major lymphoid and myeloid subpopulations of *R26-ILEI^ind* mice receiving Dox compared to age matched *R26-ILEI^ind* mice without Dox or *ILEI^ind* mice with Dox, with the exception of a reduction in the CD11b+ subpopulation (Fig 3C). This suggests that ILEI overexpression was not stimulating an immune response. On the other hand, hemoglobin concentration was significantly decreased in *R26-ILEI^ind* mice receiving Dox at 3, 2, and 1 weeks before death compared to age matched mice in the *ILEI^ind* with Dox and *R26-ILEI^ind* without Dox groups (all $p<0.05$) (Fig 3D). RBC counts were also significantly decreased one week prior to death compared to the control groups (both $p<0.01$), while HCT values were significantly decreased from 2 weeks prior to death ($p<0.05$) (Fig 3D). These reduced parameters were accompanied with reduced mean corpuscular volume (MCV), mean corpuscular hemoglobin (MCH), and mean corpuscular hemoglobin concentration (MCHC) values, indicative of microcytic hypochromic anemia.

To investigate whether any of these effects were direct consequences of ILEI overexpression and thus, reversible upon turning off the transgene, freshly weaned mice were treated with Dox water from week 3 to 13, followed by 2 weeks of Dox withdrawal and were weekly monitored for blood parameters. As shown in Fig 4A, ILEI overexpressing mice showed lower body weight over control mice, and they did not recover from the retarded growth during the two weeks after the transgene was switched off ($p<0.01$). Importantly, hemoglobin significantly decreased after 3 weeks of ILEI induction, followed by a drop in RBC and HCT levels with a shift of 3 weeks accompanied with significantly reduced MCHC levels, whereas MCV reduction, a sign of microcytic anemia, was not consistent, most likely having been masked by the interfering size reduction of erythrocytes during postnatal development until adulthood [38]. All parameters recovered to similar levels of the control groups one week after dox withdrawal (Fig 4B). This suggests that the anemia was tightly linked to ILEI overexpression and was rapidly reversible.

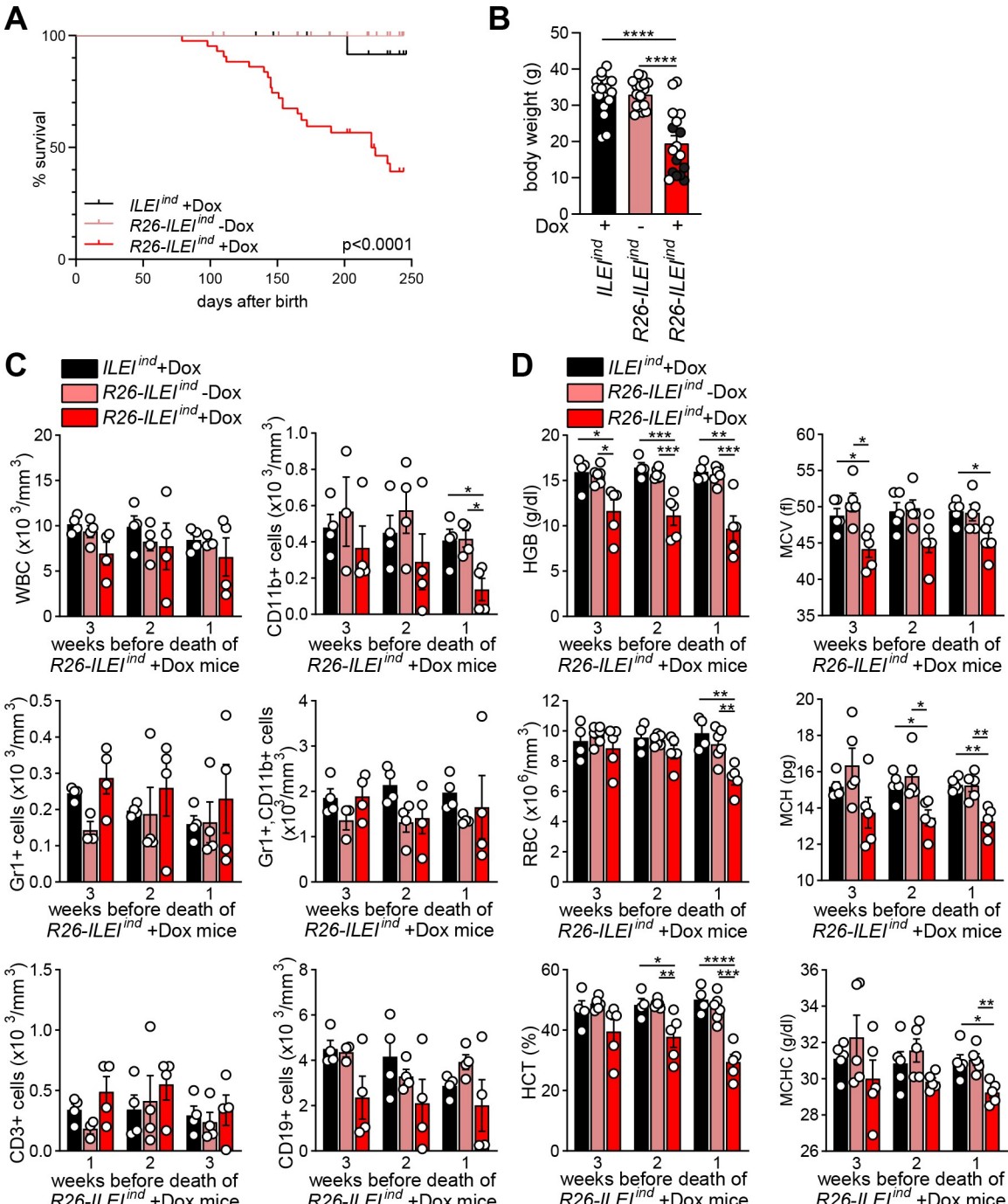

**Fig 3. Inducible ILEI overexpression results in reduced life span, body weight, and microcytic hypochromic anemia.** (A) Survival plot of *ILEI*<sup>ind</sup> and *R26-ILEI*<sup>ind</sup> mice kept on normal or switched to Dox drinking water at 3 weeks of age. (B) Mean body weight ± SEM of *R26-ILEI*<sup>ind</sup> mice kept on Dox drinking water measured one week before death (dark data points) or at experiment endpoint (white data points) compared to age-matched mice used as Dox treatment and genetic controls. (C) Mean number of WBC, CD11b+, Gr1+, CD11b and Gr1 double-positive, CD3+ and CD19+ cells ± SEM measured at last three weeks before death of *R26-ILEI*<sup>ind</sup> mice kept on Dox water and compared to age-matched mice used as Dox treatment and genetic controls. (D) Mean HGB concentration (top left), RBC count (middle left) HCT percentage (bottom left), MCV (top right), MCH (middle right) and MCHC (bottom right) ± SEM measured at last three weeks before death of *R26-ILEI*<sup>ind</sup> mice kept on Dox water and compared to age-matched mice used as Dox treatment and genetic controls. (B-D) Statistical significance was determined by one-way ANOVA and is marked with asterisks (*$p<0.05$; **$p<0.01$; ***$p<0.001$; ****$p<0.0001$*).

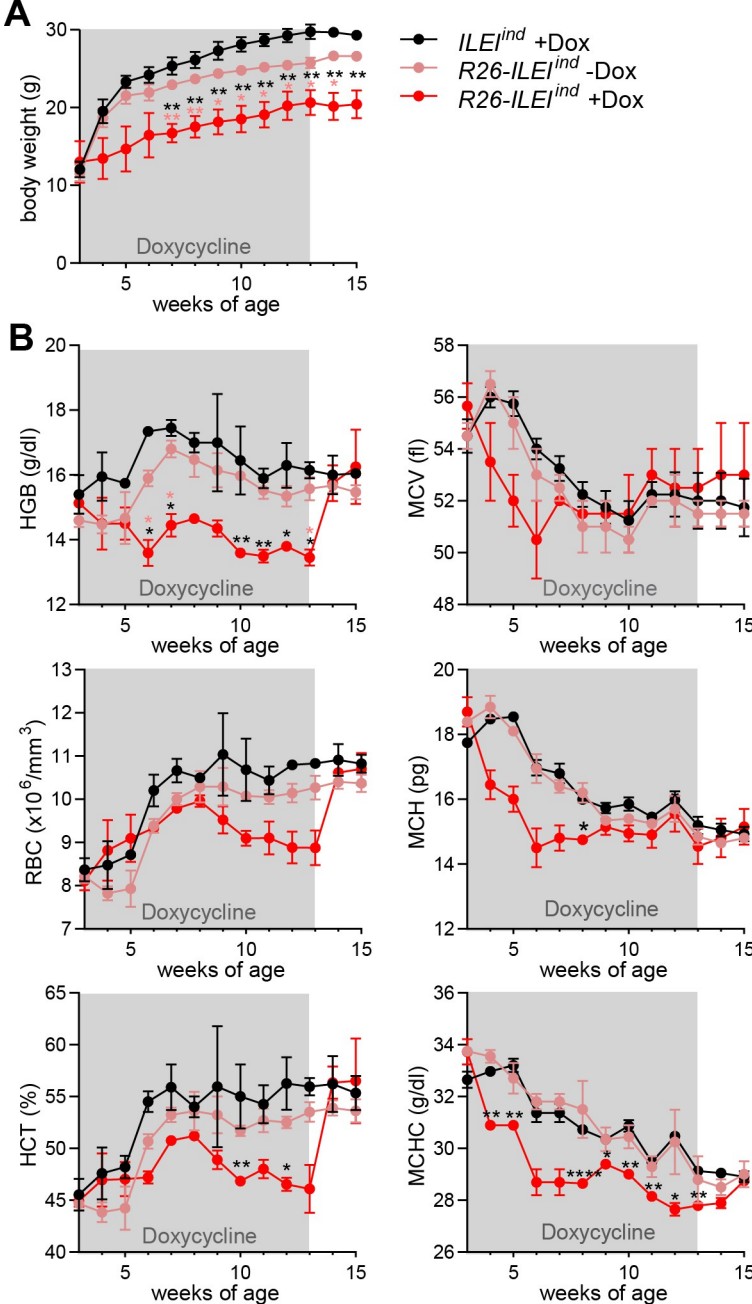

**Fig 4. Anemic phenotype is reversible upon withdrawal of temporary ILEI overexpression.** (A, B) Mean (A) body weight and (B) HGB concentration (top left), RBC count (middle left), HCT percentage (bottom left), MCV (top right), MCH (middle right) and MCHC (bottom right) ± SEM of *R26-ILEI*^*ind*^ mice kept temporarily on Dox water from week 3 to 13 of age and measured weekly until week 15 of age compared to littermates used as Dox treatment and genetic controls. (A, B) Statistical significance was determined by one-way ANOVA and is marked with asterisks (*$p < 0.05$; **$p < 0.01$; ***$p < 0.001$; ****$p < 0.0001$).

## *ILEI* overexpression in hematopoietic cells does not influence overall fitness and red blood cell parameters

In order to investigate if anemia was caused by a malfunction in the erythroid lineage due to ILEI overexpression, we next examined the effects of *Vav1*-driven overexpression of the *Fam3c/ILEI* transgene on overall fitness and blood parameters. Vav1 is exclusively expressed in all hematopoietic cells, so this provides tissue specific expression to contrast with the ubiquitous expression driven by Rosa26. Western blot analysis demonstrated that ILEI-FLAG was expressed in the bone marrow of *Vav-ILEI*[ind] mice upon Dox administration via the drinking water or diet and appeared dose dependent (Fig 5A). Importantly, overexpression of ILEI-FLAG by Dox induction in the hematopoietic compartment did not alter the body weight of the mice even after over 200 days of Dox in their diet (Fig 5B). Flow cytometry analysis of the spleen with focus on myeloid leukocyte populations revealed that ILEI overexpression led to reduced number of eosinophils (Fig 5C), indicating that the reduced number of CD11b+ myeloid cells observed in the *R26-ILEI*[ind] ubiquitous overexpression model might be a consequence of intrinsic ILEI-linked mechanisms in the hematopoietic system. This had, however, no influence on parameters related to anemia, the hemoglobin levels, RBC count, and HCT percentage of *Vav-ILEI*[ind] mice kept temporarily on Dox diet were similar compared to littermates used as Dox treatment and genetic controls (Fig 5D). These data demonstrate that no intrinsic mechanisms of erythroid development were dysregulated by ILEI and that hematopoietic ILEI hyperfunction despite of an effect on myeloid lineage composition did not contribute to the morbidity and mortality of the mice.

## Ubiquitous ILEI overexpression leads to reduced iron levels in the serum and in hepatocytes, and to liver damage

Next, to further understand the anemic effects of ubiquitous ILEI overexpression we analyzed the iron levels of the transgenic mice. The serum iron levels were significantly reduced in *R26-ILEI*[ind] mice on Dox compared to control *R26-ILEI*[ind] mice without Dox ($p < 0.01$) and *ILEI*[ind] mice on Dox ($p < 0.05$) (Fig 6A). As the liver is the main organ involved in iron metabolism, we also undertook analysis for serum markers of liver function: ALT and AST. ALT (Fig 6B) was significantly increased in *R26-ILEI*[ind] mice on Dox drinking water relative to the mice without Dox ($p < 0.05$) and genetic control mice (*ILEI*[ind], $p < 0.01$), and AST levels were also increased (Fig 6C) in *R26-ILEI*[ind] mice on Dox relative to the mice without Dox ($p < 0.05$). Though mice overexpressing ILEI already had a lower body weight, the liver-to-body weight ratio (Fig 6D) was additionally significantly lower in these animals compared to Dox treatment and genetic controls ($p < 0.001$ and $p < 0.05$, respectively), also represented on macroscopic images in Fig 6E. Furthermore, all these observations had an exaggerated penetrance in moribund ILEI overexpressing mice compared to mice of the same group that survived to experimental endpoint (dark vs. white data points in Figs 3B and 6B–6D), indicating that beside an overall retarded growth and lower fitness Dox-induced *R26-ILEI*[ind] mice had specific problems in liver function and size maintenance that were linked to death. There were also some evident differences in histological analysis of liver sections. H&E staining suggested that hepatocytes were smaller with small nuclei and the liver lost its typical hexagonal lobular structure with intermitting central vein and portal venule arrangements in mice expressing ILEI-FLAG compared to the two control groups (Fig 6F). This was alongside significantly decreased numbers of cells positive for Ki67, a proliferation marker ($p < 0.05$ vs. Dox control, Fig 6G and 6H), and increased rates of activated caspase 3 positive cells, indicative of apoptosis, in *R26-ILEI*[ind] mice on Dox ($p < 0.001$ vs. Dox control and $p < 0.0001$ vs. genetic control, Fig 6I and 6J). Prussian blue staining showed lower number of hepatocytes with ferric iron ($Fe^{3+}$) storage in the liver

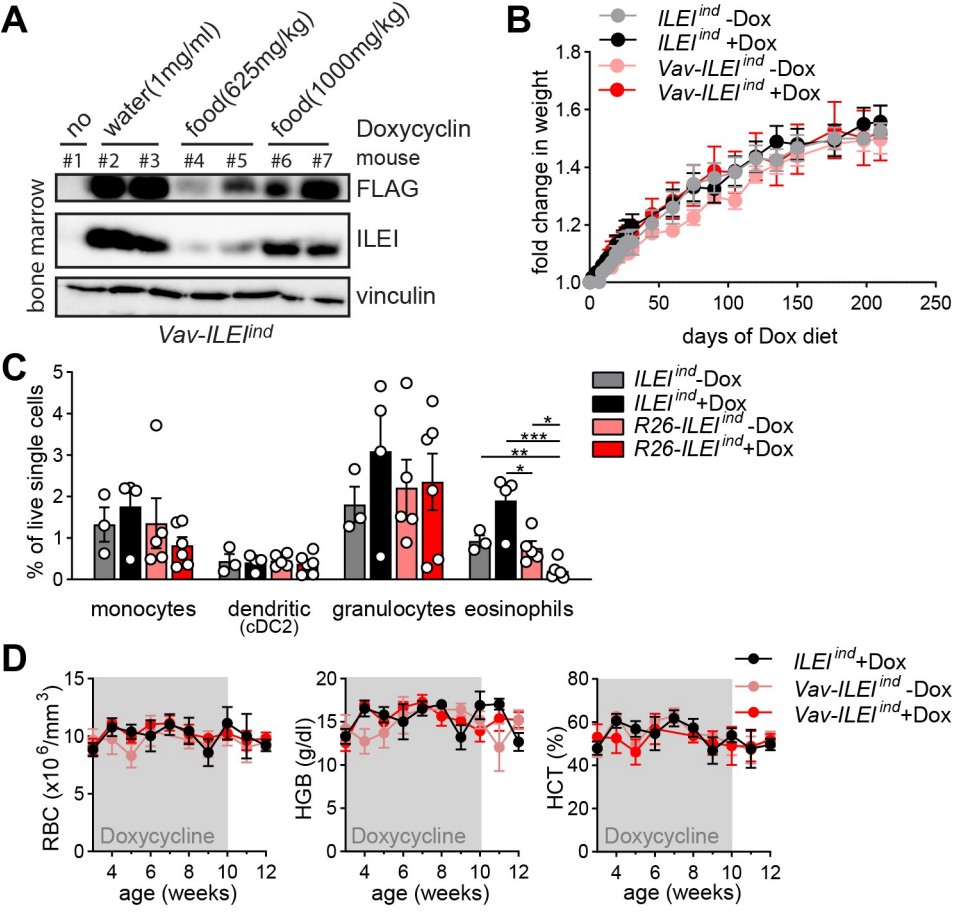

**Fig 5. ILEI overexpression in hematopoietic cells does not influence overall fitness and red blood cell parameters.**
(A) Western blot analysis of freshly isolated bone marrow from dual-transgenic *Vav-ILEI*[ind] mice kept on Dox supplemented either in the drinking water or in the food at different concentrations for 3 days. FLAG and ILEI antibodies were used for transgene detection, vinculin as loading control. (B) Mean body weight ± SEM of *ILEI*[ind] and *Vav-ILEI*[ind] mice kept on normal or Dox diet. (C) Mean percentage of live cells ± SEM of listed myeloid subpopulations in the spleen of *ILEI*[ind] and *Vav-ILEI*[ind] mice kept on normal or Dox diet. (D) Mean HGB concentration, RBC count and HCT percentage ± SEM of *Vav-ILEI*[ind] mice kept temporarily on Dox diet from week 3 to 10 of age and measured weekly until week 12 of age compared to littermates used as Dox treatment and genetic controls. Statistical significance was determined by student's *t*-test and is marked with asterisks (*$p<0.05$; **$p<0.01$; ***$p<0.001$).

tissue of ILEI-FLAG expressing mice compared to the controls (p< 0.05, Fig 6K and 6L). Therefore, these results suggest that the reduced serum iron levels observed upon ILEI overexpression were linked with reduced iron content in the liver. In addition, the liver showed malfunction and reduced hepatoprotection, which in combination with the reduced iron levels may be indicative of reduced iron uptake and heme production by hepatocytes, contributing to anemia.

## Mice with induced ubiquitous ILEI overexpression develop liver fibrosis

The apparent liver dysfunction showed signs indicative of liver fibrosis, thus, fibrosis markers were analyzed histologically as shown in Fig 7. Immunohistochemistry with anti-fibronectin antibodies showed that *R26-ILEI*[ind] mice on Dox had significantly higher rates of stained areas compared to *R26-ILEI*[ind] mice without Dox and *ILEI*[ind] mice who received Dox (both

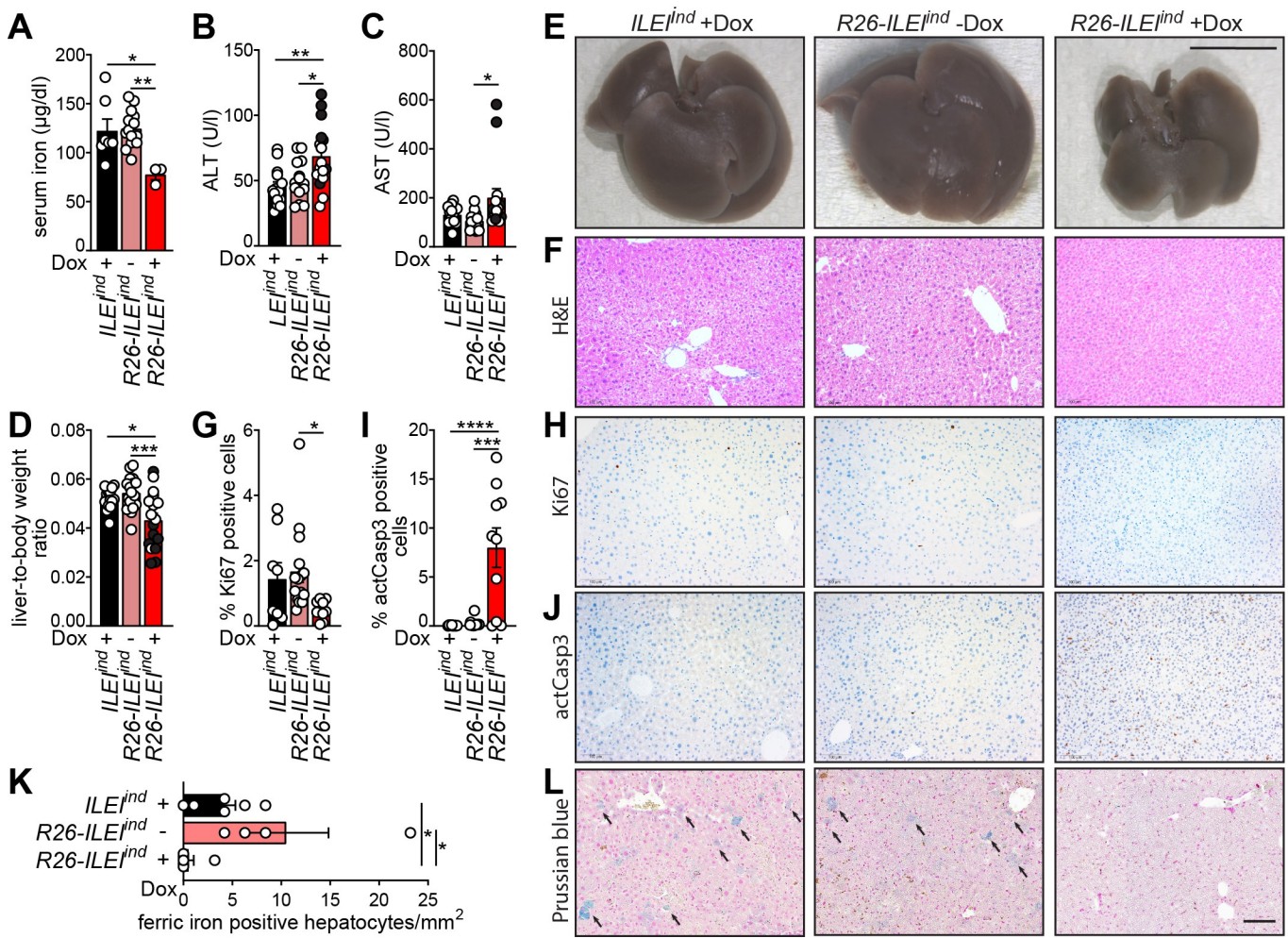

**Fig 6. Ubiquitous ILEI overexpression leads to reduced iron levels in the serum and in hepatocytes and to liver dysfunction.** (A) Mean serum iron levels ± SEM of *R26-ILEI^ind^* mice kept on Dox drinking water measured one week before death compared to age-matched mice used as Dox treatment and genetic controls. (B-C) Mean (B) ALT and (C) AST levels ± SEM of *R26-ILEI^ind^* mice kept on Dox drinking water measured one week before death (dark data points) or at experiment endpoint (white data points) compared to age-matched mice used as Dox treatment and genetic controls. (D) Mean liver-to-body weight ratio ± SEM of *R26-ILEI^ind^* mice kept on Dox drinking water measured one week before death (dark data points) or at experiment endpoint (white data points) compared to age-matched mice used as Dox treatment and genetic controls. (E-F) Representative (E) macroscopic images and (F) H&E stained thin sections of the liver of moribund *R26-ILEI^ind^* mice kept on Dox drinking water compared to age-matched mice used as Dox treatment and genetic controls. (G-L) Mean percentage of (G) Ki67 and (I) activated Caspase 3 (actCasp3) positive cells ± SEM with representative images of corresponding (H) Ki67 and (J) actCasp3 immunohistochemistry of the liver of moribund *R26-ILEI^ind^* mice kept on Dox drinking water compared to age-matched mice used as Dox treatment and genetic controls. (K,L) Iron content of the liver shown as (K) Fe3 + positive hepatocytes per mm² ± SEM and (L) representative images of moribund *R26-ILEI^ind^* mice kept on Dox drinking water compared to age-matched mice used as Dox treatment and genetic controls. Scale bars, (E) 1 cm, (F,H,J,L) 100μm. Statistical significance was determined by one-way ANOVA (A,B,C,D,G,I) or student's *t*-tests (K) and marked with asterisks (*$p<0.05$; **$p<0.01$; ***$p<0.001$; ****$p<0.0001$).

$p<0.0001$, Fig 7A and 7C). Similar results were found with S100A4 immunohistochemistry (both $p<0.05$, Fig 7B and 7D). Furthermore, *R26-ILEI^ind^* mice on Dox were highly enriched for areas of disordered and fibrous collagen deposition compared to the two control groups that had an overall weak collagen signal with the exception of accumulations around blood vessels as visualized by Sirius Picric Red staining (Fig 7E). Similarly, *R26-ILEI^ind^* mice on Dox showed a more dense appearance of bile ducts and an upregulation of E-cadherin expression in these structures that was not evident in the two control groups of mice by immunohistochemistry with anti-E-cadherin antibodies (Fig 7F). These data support the view that liver

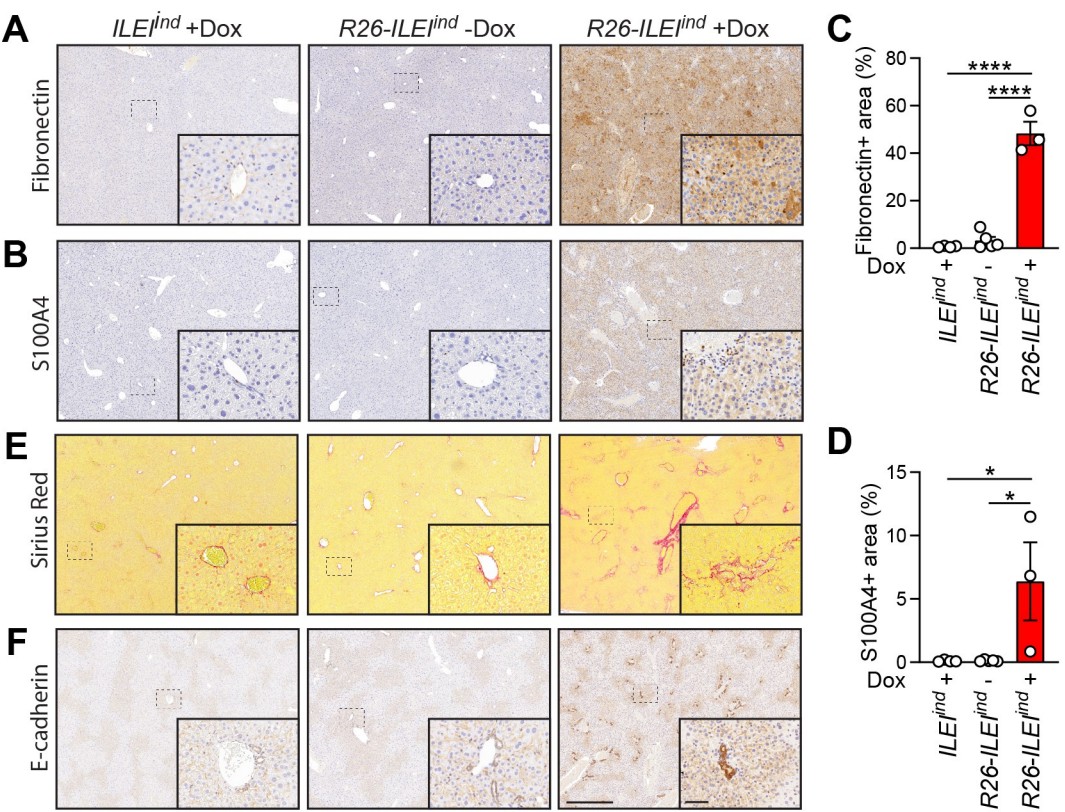

**Fig 7. Mice with induced ubiquitous ILEI overexpression develop liver fibrosis.** (A-D) Representative images of (A) Fibronectin and (B) S100A4 immunohistochemistry and their quantifications shown as mean percentage of (C) Fibronectin and (D) S100A4 positive area ± SEM of the liver of moribund *R26-ILEI^{ind}* mice kept on Dox drinking water compared to age-matched mice used as Dox treatment and genetic controls. (E-F) Increased collagen deposition and bile duct density shown by (E) Sirius Picric Red staining and (F) E-cadherin immunohistochemistry. Scale bar, 200μm; scale bar for inlets, 50μm. (B, C) Statistical significance was determined by one-way ANOVA and marked with asterisks (*$p<0.05$; ****$p<0.0001$).

fibrosis results from overexpression of ILEI in the liver. This could in turn lead to severe liver dysfunction and eventually to death.

## Discussion

In this study we present the development of a mouse model to study the effects of elevated ILEI expression *in vivo*. We demonstrate the successful generation and establishment of a transgenic mouse line with inducible ubiquitous ILEI overexpression (*R26-ILEI^{ind}* mice). The *R26-ILEI^{ind}* mice have the *M2rtTA* inserted into the *Rosa26* locus and the Dox-dependent tetO:*ILEI-FLAG* construct in the 3'UTR region of the *Col1a1* locus, resulting in ubiquitous inducible, Rosa26 promoter-driven expression of M2rtTA which recognizes the Tet-operon (tetO) only in the presence of Dox to control the expression of the *Fam3c/ILEI-FLAG* transgene. The Dox induction showed dose dependency. Mice with the Dox-dependent *ILEI-FLAG* construct at the *Col1a1* locus without the *M2rtTA* inserted at the *Rosa26* locus were used as genetic control and showed no measurable expression of ILEI-FLAG upon Dox administration. The Tet-On system used in this model also allows tissue specific expression. In this case we used the rtTA under the control of the Vav1 promoter, which is exclusively active in hematopoietic cells [39], to limit the overexpression of ILEI to those cell types. Therefore, this novel Tet-On inducible *Fam3c/ILEI* transgenic mouse strain will provide a useful model for

tissue specific and timely controlled overexpression of ILEI and by this represents a versatile tool to model the effect of elevated ILEI expression in diverse disease conditions, including cancer.

As ILEI is vital in the EMT process, overexpression of ILEI early in development could not be excluded from interfering with mouse viability in a drastic if not lethal manner. Therefore, we started Dox induction at weaning age. ILEI-FLAG was expressed at high levels in most tissues examined, suggesting that the transgenic mice were ubiquitously expressing the fusion protein. Characterization of the mouse phenotype on induction of ILEI overexpression showed a shorter lifespan, reduced body weight, microcytic hypochromic anemia, and liver dysfunction and fibrosis compared to control mice. However, the anemia was quickly reversible at a young age upon withdrawal of ILEI induction. While this is the first study to develop an inducible ubiquitous overexpression of ILEI, previous mouse models have been established that involved full knock-out of ILEI expression or neuron specific overexpression of ILEI. Those studies were investigating specific effects of ILEI in influencing bone mineral density or Alzheimer's disease. They found very small effects on the whole organism [18–20]). Interestingly, one of the *Fam3c* knock-out studies, alongside their main interest in bone morphology, found male mice showing hematological changes not evident in females [19]. As our study was not intended to investigate a role of ILEI in bone we did not undertake a detailed analysis of the bones of the transgenic mice. This will obviously be an interesting study in the future alongside a detailed examination of other important organs such as the heart, skeletal muscles, and brain. However, looking at the hematological changes of the previous study in detail reveals that male mice with *Fam3c* knock-out had elevated numbers of polymorphonuclear neutrophils and reduced amounts of lymphocytes, RBC counts were higher with slightly smaller corpuscular volume of the cells, total hemoglobin did not differ significantly [19]. This reasonably supports the hematological changes and the role of ILEI in anemia found in our study and suggests that in future it might be relevant to examine gender differences.

To understand the cause of the anemia in mice ubiquitously expressing ILEI we utilized the ability of the Tet-On system to be switched to tissue specific expression. Hematopoietic cells undergo proliferation and differentiation events to produce mature blood cells throughout the lifespan of the individual [40]. Overexpression of the *Fam3c/ILEI* transgene in hematopoietic cells alone did not render mice anemic or lower overall fitness. These results suggest that ILEI overexpression was not dysregulating intrinsic mechanisms of erythroid development and that overdriving of hematopoietic ILEI function did not contribute to the mortality of the mice. Ubiquitous induction of ILEI overexpression reduced serum iron levels, so this was indicative of a malfunction in iron absorption, cellular uptake, or heme production. Iron deficiency anemia normally presents as microcytic hypochromic anemia where RBCs have a smaller mean corpuscular volume (MCV) than normal. In accordance with this, mice in this study also had reduced MCV, indicative of microcytic hypochromic anemia caused by iron deficiency.

As the liver plays the major role in controlling the systemic iron balance within the body [41], we next examined the effect of ILEI overexpression in the liver. Our results show that the liver was severely affected with increased ALT and AST levels, reduced liver size, increased apoptosis, and reduced ferric iron content of hepatocytes. This dysfunction was easily visualized in histological sections having a fibrotic appearance with increased areas of fibronectin and S100A4 compared to control sections, and obvious collagen bundles. Meanwhile detection of E-cadherin showed areas of intense staining around bile ducts that was not seen in the control samples. This is reminiscent of the enhanced hepatic E-cadherin production in mouse bile duct ligation models of cholestatic fibrosis [42, 43]. These data indicate that high ILEI expression in the liver might reduce hepatoprotection and induce liver fibrosis, which leads to liver dysfunction, disturbed iron metabolism and eventually to death. This result needs careful

consideration by researchers interested in the role ILEI plays in metabolic diseases. In particular, diabetes and non-alcoholic liver disease models suggest overexpression of ILEI as potential therapy to overcome glucose intolerance, insulin resistance and liver steatosis linked to reduced ILEI expression in the liver [44, 45]. However, our data suggest that therapeutic approaches involving overexpression of ILEI would have to be very carefully controlled because they might result in liver fibrosis. Thus, our model can serve as a suitable tool to investigate long term effects of the described short-term and *in vitro* observations, and our data indicate that ILEI overexpression might be linked to unexpected risks in liver function.

The livers of mice overexpressing ILEI have many similar characteristics to liver cirrhosis in human disease, with evidence of liver dysfunction alongside increased levels of fibrosis markers and fibrous and disordered collagen. Cirrhosis is an advanced stage of liver fibrosis accompanied by distortion of the hepatic vasculature, which results in compromised liver function [46]. Late-stage cirrhosis often results in smaller liver volume, as seen in this study, and this may be related to poor prognosis of patients with hepatocellular disease [47]. On the other hand, cirrhosis usually develops after a sustained period of inflammation [46]. In this study there was no evidence of a period of inflammation or hepatitis from ILEI overexpression. Similarly, while cirrhosis increases the risk of liver cancer [46], we did not observe any sign of malignant transformation in the liver. Overall, ILEI overexpression did not cause spontaneous tumor formation in any organs as analyzed by autopsy of moribund and endpoint mice.

This study raises the issue of low levels of iron in the liver and liver dysfunction. However, most investigations of iron and liver damage highlight the role of high levels of iron being a cause of oxidative damage resulting in liver fibrosis [48–50]. In this situation with low iron levels in the liver, blood analysis indicated few effects on WBC counts or major lymphoid and myeloid subpopulations suggesting that ILEI overexpression was not stimulating an immune response. It seems more likely from our results that overexpression of ILEI in the liver results in liver dysfunction and as a consequence iron metabolism is disturbed. This is supported by the results from specific overexpression of ILEI in hematopoietic cells. However, it is also possible that low iron levels are primary and iron deficiency is initially not tightly linked to liver dysfunction. Detailed analysis of this likely complicated mechanism is beyond the initial characterization of the mouse model presented here. Therefore, the mechanisms behind iron malabsorption indicative in this mouse model need to be investigated in more detail in the future. Those studies will involve uncoupling the effects of overexpression of ILEI in the liver from other systemic effects on iron levels and will require detailed analysis of various tissue specific ILEI expression profiles, including the intestine as main place of iron absorption. It is also important to fully understand the microcytic hypochromic anemia discovered in the mice, including analysis of the factors involved, such as transferrin and ferritin, to give a more precise understanding of its nature.

## Conclusion

The study describes the generation and initial characterization of a transgenic mouse with inducible expression of ILEI. Ubiquitous induction of ILEI overexpression at weaning age resulted in mice with a shortened lifespan, reduced body weight, and microcytic hypochromic anemia. These characteristics seemed to be related to high levels of ILEI expression in the liver, which showed dysfunction and fibrosis. Therefore, high ILEI expression in the liver might reduce hepatoprotection and induce liver fibrosis. This could in turn lead to liver dysfunction, disturbed iron metabolism, and eventually to death.

## Supporting information

**S1 File.**
(ZIP)

## Acknowledgments

We thank Nicole Leitner and Simone Müller for their involvement in the generation of the transgenic mouse line and in the speed congenic procedure, Gabriella Litos for assisting at genotyping, Boris Kovacic for experimental guidance in Flow Cytometry and Gergely Szakacs, Robert Eferl and Lukas Kenner for constructive discussions and critical reading of the manuscript. The mouse strain *VavrtTA* was kindly provided by Michael Bader and Ross Dickins. CRediT (Contributor Roles Taxonomy) was used to define contributor roles of the authors. Melanie Colegrave from Molecular Cell Research provided writing assistance.

## Author Contributions

**Conceptualization:** Agnes Csiszar.

**Data curation:** Martin Holcmann.

**Formal analysis:** Martin Holcmann, Veronica Moreno-Viedma, Bernhard Robl, Carina Mühlberger, Dagmar Gotthardt.

**Funding acquisition:** Agnes Csiszar.

**Investigation:** Ulrike Schmidt, Betül Uluca, Iva Vokic, Barizah Malik, Thomas Kolbe, Caroline Lassnig.

**Resources:** Thomas Kolbe, Caroline Lassnig, Maria Sibilia, Thomas Rülicke, Mathias Müller, Agnes Csiszar.

**Supervision:** Maria Sibilia, Thomas Rülicke, Mathias Müller, Agnes Csiszar.

**Writing – original draft:** Agnes Csiszar.

**Writing – review & editing:** Ulrike Schmidt, Betül Uluca, Iva Vokic, Barizah Malik, Thomas Kolbe, Caroline Lassnig, Veronica Moreno-Viedma, Bernhard Robl, Carina Mühlberger, Dagmar Gotthardt, Maria Sibilia, Thomas Rülicke, Mathias Müller.

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
