## [Decision Letter · Decision Letter 0]

16 Feb 2023

PONE-D-23-00773Inducible overexpression of a FAM3C/ILEI transgene has pleiotropic effects with shortened life span, liver fibrosis and anemia in micePLOS ONE

Dear Dr. csiszar,

Thank you for submitting your manuscript to PLOS ONE. After careful consideration, we feel that it has merit but does not fully meet PLOS ONE’s publication criteria as it currently stands. Therefore, we invite you to submit a revised version of the manuscript that addresses the points raised during the review process.

 Please consider and address, point by point, reviewer #2's objections and recommendations.  While major new studies (recommended by the reviwer) may not be necessary, clearly indicate why that is so. 

We look forward to receiving your revised manuscript.

Kind regards,

Michael Klymkowsky, Ph.D.

Academic Editor

PLOS ONE

Journal Requirements:

"The study was supported by a grant of the Austrian Science Fund (FWF) ZK-81B (DG and CM), the Overseas Scholarship of the University of the Punjab, Pakistan (BM) and the Fellinger Krebsforschung (AC)."

"The study was supported by a grant of the Austrian Science Fund (FWF) ZK-81B (DG and CM), the Overseas Scholarship of the University of the Punjab, Pakistan (BM) and the Fellinger Krebsforschung (AC)."

"The study was supported by a grant of the Austrian Science Fund (FWF) ZK-81B (DG and CM), the Overseas Scholarship of the University of the Punjab, Pakistan (BM) and the Fellinger Krebsforschung (AC)."

Reviewers' comments:

Reviewer's Responses to Questions

**Comments to the Author**

1. Is the manuscript technically sound, and do the data support the conclusions?

Reviewer #1: Yes

Reviewer #2: Partly

2. Has the statistical analysis been performed appropriately and rigorously? 

Reviewer #1: Yes

Reviewer #2: Yes

3. Have the authors made all data underlying the findings in their manuscript fully available?

Reviewer #1: Yes

Reviewer #2: Yes

4. Is the manuscript presented in an intelligible fashion and written in standard English?

Reviewer #1: Yes

Reviewer #2: Yes

5. Review Comments to the Author

Reviewer #1: The manuscript PONE_D_23-00773 by Schmidt et al. is very interesting indicating tissue-specific effects for ILEI overexpression. The manuscript is well written and the data presented support the conclusions.

Reviewer #2: In this study, the authors generated and characterized a Tet-ON inducible ILEI-transgenic mouse strain. They found that mice with induced ubiquitous ILEI-FLAG overexpression at weaning age exhibited a shortened lifespan, reduced body weight, iron-deficiency anemia, and liver fibrosis with dysfunction. Reduced iron levels were observed in the serum and in hepatocytes.

This study is an initial and incomplete characterization of Tet-ON inducible ILEI-transgenic mice. This reviewer has the following comments on this manuscript.

1) Using quantitative methods such as immunoblotting and ELISA, the authors should show expression levels of ILEI in major organs including the brain, heart, lung, liver, spleen, kidney, skeletal muscles, and bone marrow of induced ILEI-Tg mice used in this study as comparing with those of control and wild-type mice.

2) Previously, the author's group reported that homodimers of ILEI are exclusively active (Krai M, 2017; Jansson AM, 2017). They should examine whether overexpressed ILEI-FLAG in their mice is dimerized.

3) Iron deficiency anemia is usually microcytic hypochromic but not normocytic hypochromic. The authors should show MCV and MCH (or MCHC) values. In addition, effectiveness to iron administration can indicate whether anemia of ILEI-overexpressed mice is caused by iron deficiency.

4) Transferrin plays a critical role in transfer and metabolism of iron. The authors should measure serum transferrin levels of mice with induced ILEI overexpression.

5) Livers of induced ILEI Tg mice were reduced in size, showed increased apoptosis, reduced cellular iron content, and had a fibrotic phenotype. Which finding appeared firstly? Were these findings reversible upon withdrawal of ILEI induction?

6) Livers of mice with induced ILEI overexpression were somewhat reminiscent of liver cirrhosis in human. The authors should discuss about similarity and difference between these conditions.

7) Previous studies suggested that ILEI overexpression caused carcinogenesis. The authors should evaluate frequency of spontaneous cancer development after induction of ILEI overexpression.

8) In line 370 on page 17; "Figure F" ?

9) The authors refer to the same mouse line as "Fam3c transgenic" and "ILEI transgenic".

6. PLOS authors have the option to publish the peer review history of their article (what does this mean?). If published, this will include your full peer review and any attached files.

Reviewer #1: No

Reviewer #2: No

---

## [Author Response · Author response to Decision Letter 0]

12 Apr 2023

our point-by-point response to the Reviewers is uploaded as a file "Response to Reviewers"

---

## [Decision Letter · Decision Letter 1]

12 May 2023

Inducible overexpression of a FAM3C/ILEI transgene has pleiotropic effects with shortened life span, liver fibrosis and anemia in mice

PONE-D-23-00773R1

Dear Dr. csiszar,

We’re pleased to inform you that your manuscript has been judged scientifically suitable for publication and will be formally accepted for publication once it meets all outstanding technical requirements.

Kind regards,

Michael Klymkowsky, Ph.D.

Academic Editor

PLOS ONE

Additional Editor Comments (optional):

Reviewers' comments:

Reviewer's Responses to Questions

**Comments to the Author**

1. If the authors have adequately addressed your comments raised in a previous round of review and you feel that this manuscript is now acceptable for publication, you may indicate that here to bypass the “Comments to the Author” section, enter your conflict of interest statement in the “Confidential to Editor” section, and submit your "Accept" recommendation.

Reviewer #2: (No Response)

2. Is the manuscript technically sound, and do the data support the conclusions?

Reviewer #2: Yes

3. Has the statistical analysis been performed appropriately and rigorously? 

Reviewer #2: Yes

4. Have the authors made all data underlying the findings in their manuscript fully available?

Reviewer #2: Yes

5. Is the manuscript presented in an intelligible fashion and written in standard English?

Reviewer #2: Yes

6. Review Comments to the Author

Reviewer #2: (No Response)

7. PLOS authors have the option to publish the peer review history of their article (what does this mean?). If published, this will include your full peer review and any attached files.

Reviewer #2: No

---

## [Editor Report · Acceptance letter]

16 May 2023

PONE-D-23-00773R1 

Inducible overexpression of a *FAM3C/ILEI* transgene has pleiotropic effects with shortened life span, liver fibrosis and anemia in mice 

Dear Dr. Csiszar:

I'm pleased to inform you that your manuscript has been deemed suitable for publication in PLOS ONE. Congratulations! Your manuscript is now with our production department. 

Kind regards, 

on behalf of

Dr. Michael Klymkowsky 

Academic Editor

PLOS ONE